

**Impact of different solar EUV proxies and Ap index on hmF2 trend analysis**
Trinidad Duran[1,2], Bruno S. Zossi[3,4], Yamila Melendi[1,2,5], Blas F. de Haro Barbas[3,4],
Fernando S. Buezas[1,2], and Ana G. Elias[3,4]
(1) Departamento de Física, Universidad Nacional del Sur (UNS), Bahía Blanca, Argentina
(2) Instituto de Física del Sur (CONICET-UNS), Bahía Blanca, Argentina
(3) Laboratorio de Ionosfera, Atmosfera Neutra y Magnetosfera (LIANM), Facultad de
Ciencias Exactas y Tecnología (FACET), Universidad Nacional de Tucumán (UNT),
Argentina
(4) Instituto de Física del Noroeste Argentino (CONICET-UNT), Argentina
(5) Tucumán Space Weather Center (TSWC), Tucuman, Argentina
Corresponding author:
Trinidad Duran
E-mail: tduran@ifisur-conicet.gob.ar; trinidad.duran.94@gmail.com
**Abstract**
Long-term trend estimation in the peak height of the F2 layer, hmF2, needs the previous
filtering of much stronger natural variations such as those linked to the diurnal, seasonal, and
solar activity cycles. If not filtered, they need to be included in the model used to estimate
the trend. The same happens with the maximum ionospheric electron density that occurs in
this layer, NmF2, usually analyzed through the F2 layer critical frequency, foF2. While
diurnal and seasonal variations can be easily managed, filtering the effects of solar activity
presents more challenges, as does the influence of geomagnetic activity. However, recent
decades have shown that geomagnetic activity may not significantly impact trend
assessments. On the other hand, the choice of solar activity proxies for filtering has been
shown to influence trend values in foF2, potentially altering even the trend's sign. This study
examines the impact of different solar activity proxies on hmF2 trend estimations, using data
updated to 2022, including the ascending phase of solar cycle 25, and explores the effect of
including the Ap index as a filtering factor. The results obtained, based on two mid-latitude
stations, are also comparatively analyzed to those obtained for foF2. The main findings
indicate that the squared correlation coefficient, $r^2$, between hmF2 and solar proxies,
regardless of the model used or the inclusion of the Ap index, is consistently lower than in
the corresponding foF2 cases. This lower $r^2$ value in hmF2 suggests a greater amount of
unexplained variance, indicating that there is significant room for improvement in these
models. However, in terms of trend values, foF2 shows greater variability depending on the
proxy used, whereas the inclusion or exclusion of the Ap index does not significantly affect
these trends. This suggests that foF2 trends are more sensitive to the choice of solar activity





proxy. In contrast, hmF2 trends, while generally negative, exhibit greater stability than foF2 trends.

**Keywords**

Solar activity proxy, hmF2, ionosphere long-term trends, F10.7, F30, greenhouse gas increase.

**Highlights**

1. Long-term trends in hmF2 change with the solar activity proxy used for filtering but are mostly negative.

2. SN yields the weakest negative hmF2 trends, which are still negative, while foF2 trends are mostly positive.

3. Yearly hmF2 values show a linear relationship with solar proxies but improve with the inclusion of a squared term and the Ap index.

**1. Introduction**

Long-term trends in the Earth's ionosphere expected from the increase in greenhouse gas concentration along the last decades has been a topic of growing interest since the late 1980's (Roble and Dickinson, 1989; Rishbeth, 1990) with many results already published (Laštovička et al. 2012, 2014; Laštovička 2017, 2021a). It has been mainly studied through the analyses of the critical frequency of the F2 layer, foF2, that is a measure of the ionospheric peak electron density, NmF2 (=$1.24 \cdot 10^{10}$ foF2$^2$, with foF2 in MHz and NmF2 in m$^{-3}$). Even though the trends in the ionosphere linked to the greenhouse effect are expected to be more clear in the ionospheric peak height, hmF2, (Rishbeth, 1990; Rishbeth and Roble, 1992) publications analyzing foF2 trend detection are by far more numerous. One reason may be that hmF2, unlike foF2, is not directly derived from ionosonde records. It can be estimated using the Shimazaki formula (Shimazaki, 1955) based on the M(3000)F2 propagation factor, which is calculated by taking the ratio of the Maximum Usable Frequency at 3000 km (MUF(3000)) to foF2, and which dates back to the same years as foF2. However, specially during daytime hours, there are systematic differences between hmF2 derived from M(3000)F2 and the true height value. A good option is systematic hmF2 deduced by real-height analysis of automatically scaled vertical incidence digisonde ionograms but these time series are available for only a few past decades.

Regarding the selection of a best solar EUV proxy to estimate trends in the F2 region, it is a problem which dates back almost to the very beginning when long-term trends in the upper atmosphere became a topical issue, but has regained critical importance during the last few years. We could speak of two epochs discussing this issue, which are before and after the occurrence of the 2008 solar minimum. Papers analyzing trends based on time series not reaching this period, deal basically with the selection between two proxies: F10.7 and SN. After the 2008 minimum epoch, studies that analyzed time series that included cycle 23 with





its minimum in ~2008, detected that not only SN, but also F10.7 was not efficient enough for filtering solar activity. As a result, indices more directly related to UV and EUV radiation came into play, such as the core-to-wing ratio of the Mg II line, and the solar Lyman α irradiance (at 121.567 nm). It can be also said that 2021, with the works by Laštovička (2021b, 2021c), is the year when a variety of solar EUV proxies are formally introduced as options to filter ionospheric parameters as a previous step in trend estimations.

Most papers used foF2 in order to determine the effect of the different proxies over the trend values, and also to decide which of them was a best EUV indicator (de Haro Barbas et al., 2021; Zossi et al., 2023; Danilov and Konstantinova, 2023; Laštovička and Burešová, 2023; Laštovička, 2024). Laštovička (2021b) incorporated foE, and Laštovička (2021c) also global TEC.

Jarvis et al. (1998) were among the first to do a solar proxy selection for estimating hmF2 trends. They specifically compared F10.7 and SN, choosing F10.7 due to its slightly smaller variance in trend estimates during solar cycles 23 and 24, which marked a period of significant discrepancy compared to earlier cycles, ending in 1995. Jarvis et al. (2002), added E10.7 to the solar proxies' options for hmF2 trend estimations, but its performance was almost identical to F10.7.

Laštovička et al. (2006), for foF2 trend analysis, compared SN to F10.7 and E107. They distinguished between adjusted and observed in the case of the last two proxies, with the observed F10.7 and E10.7 appearing to be the best correcting factors for filtering or modeling solar activity effects prior to trend estimation. Observed F10.7 performed the best also in the study of Ulich et al. (2007), analyzing foF2 trends as well, which is reasonable since the solar radiative energy reaching Earth is modulated by the variation in the Earth–Sun distance.

The idea was to provide a comprehensive overview of the evolution in the effort to select the best solar proxy for detecting long-term trends in ionospheric parameters, but the task turned out to be much larger than anticipated. This is not only due to the many years that have passed since the proxy selection issue was first identified as a conflict in the field of long-term trends, but also because the problem has become increasingly complex. On one hand, there are numerous proxies, and on the other hand, two variations in solar activity have become more apparent over the years that were not as evident with shorter data series. One is the prominence of the Gleissberg cycle in the maximum solar activity, which became clear with six complete cycles of data showing a long-term periodic modulation (~80-90 years, corresponding to the Gleissberg periodicity) and the decline of the last two minima (~2008 and ~2019) compared to previous minima. These two "trends" in solar activity are not identical in every proxy. Therefore, we will end with the review of this major issue in trend estimation here, suggesting it as a future task to be carefully revisited, and proceed directly to our analysis of this conflict with hmF2 data updated to the year 2022. The issue of including or not Ap, seemed to have a weaker effect than the solar proxy selection, but was also mostly analyzed in foF2 trend studies. So, we will focus in the two problems: the solar activity proxy selection and whether accounting for geomagnetic activity makes a difference or not in trend values, making a comparison with foF2 case.





The next three sections outline the data sets used and methodology. The results are provided
in section 5, followed by the discussion and concluding remarks in section 6.

**2. Data sets**
**2.1 Ionospheric data**
Hourly monthly medians of the ionospheric propagating factor at 3000 km of the F2 layer,
identified as M(3000)F2, and foF2 from two mid-latitude ionospheric stations were analyzed
for the period 1960-2022: Rome (41.5°N, 12.3°E) and Juliusruh (54.6°N, 13.4°E). Databases
were obtained from the World Data Centre (WDC) for Space Weather, Australia, accessible
at https://downloads.sws.bom.gov.au/wdc/iondata/au/ and from Damboldt and Suessman
database (Damboldt and Suessman, 2012) available in the same WDC
(https://downloads.sws.bom.gov.au/wdc/iondata/medians/). In the case of Rome, to extend
the dataset until 2022, additional data were incorporated from the Digital Ionogram Data
Base (DIDBase) at Lowell GIRO Data Center (LGDC), (Reinisch and Galkin, 2011). A 7-
year overlap (2001-2007) between the two datasets was examined to confirm series
homogeneity, resulting in a reasonable agreement of over 95% between the series.
Autoscaled hmF2, together with M(3000)F2, for the period 2001-2022 from the LGDC were
also used for Rome and Juliusruh to test the height formula chosen in this study. Data from
the DIDBase at LGDC has a frequency from 5 to 30 minutes. In order to obtain the monthly
medians, we first selected data with Autoscaling Confidence Score (CS) greater than 70%,
and then estimated for each month the hourly medians.
To calculate hmF2 from M(3000)F2, the Shimazaki formula was used (Shimazaki, 1955):
$$hmF2 = \frac{1490}{M(3000)F2} - 176 \qquad (1)$$
Annual mean foF2 and hmF2 values were assessed for 0 LT and 12 LT.
While the value of hmF2 depends on the formula used, and it is closer to the "real" value for
more precise ones than Equation (1), such as those given by Bradley and Dudeney (1973),
Dudeney (1974), and Bilitza et al. (1979), the trend values may not differ much. In this
regard, some studies suggest this is the case (Bremer, 1998), while others indicate that trends
values, and even the sign, may change depending on the formula used (Ulich, 2000; Jarvis et
al., 2002). We conducted a test for the two stations here analyzed described in Section 3
leading us to conclude that the Shimazaki formula is reasonable and reliable for the analysis
outlined in this research.
In the case of M(3000)F2 monthly median data for Rome, from January 1960 to December
2022, there are no missing values for the selected local times. For Juliusruh there are a total
of 8 missing values that correspond to the monthly medians of May 1977, September-
October-November 1978, October 1983, August 2009, July 2020 and January 2022, for both
local times. We considered that the mean annual values are all representative considering that



the worst case is 1978 with only three months missing. In the case of foF2 for both stations,
at 0 and 12 LT, there are no missing data in the monthly median records.

## 2.2 Solar EUV proxies and geomagnetic activity data

The five most commonly used solar EUV radiation proxies were employed together with the
geomagnetic activity Ap index. The five selected proxies are:
(1) Magnesium II core-to-wing ratio (MgII) (Snow et al., 2014) represents the ratio of the h
and k lines of the solar Mg II emission at 280 nm to the background solar continuum near
280 nm. The annual mean time series was calculated as the average of daily values from the
composite extended MgII series obtained from the University of Bremen at
https://www.iup.uni-bremen.de/UVSAT/data/.
(2) Hydrogen Lyman α flux (Fα) (Machol et al., 2019) in $W/m^2$ units that is the full disk
integrated solar irradiance over 121-122 nm, dominated by the solar HI 121.6 nm emission.
The annual mean time series was estimated as the average of daily values of the composite
series sourced from the LASP Interactive Solar Irradiance Data Center, University of
Colorado, at https://lasp.colorado.edu/data/timed_see/composite_lya/lyman_alpha_composite.nc.
(3) The revised sunspot number (SN). The annual mean values were directly obtained from
SILSO (Sunspot Index and Long-term Solar Observations - Royal Observatory of Belgium,
Brussels) accessible at http://www.sidc.be/silso/datafiles.
(4) F10.7 that is the flux density of radio emissions from the Sun at 10.7 cm wavelength
(2800 MHz) in $sfu=10^{-22}Ws/m^2$, measured at the Earth's surface. The annual time series was
estimated as the average of the monthly mean series available from Space Weather Canada
at https://spaceweather.gc.ca/forecast-prevision/solar-solaire/solarflux/sx-en.php.
(5) F30 that is the flux density of radio emissions from the Sun at 30 cm wavelength (1000
MHz), in $sfu=10^{-22}Ws/m^2$, measured at the Earth's surface. The annual mean time series was
estimated as the average of daily values provided by the Nobeyama Radio Polarimeters
(NoRP) at https://solar.nro.nao.ac.jp/norp/index.html.
The geomagnetic activity index Ap annual mean series was estimated as the average of daily
values supplied by the Kyoto World Data Center for Geomagnetism at
https://wdc.kugi.kyoto-u.ac.jp/index.html.

## 3. Testing the hmF2 Shimazaki formula for use in this analysis

The Shimazaki formula to obtain hmF2 based only on M(3000)F2 is adequate at nighttime
hours, when the ionization below the F2 region is weak. As this ionization begin to increase,
this formula systematically overestimates hmF2. This can be seen in Figure 1 where the
average of the monthly median hmF2 values along 2001-2022 is plotted in terms of month.
At 0 LT a good agreement is noticed between the autoscaled and the Shimazaki heights,
which declines in the case of 12 LT.





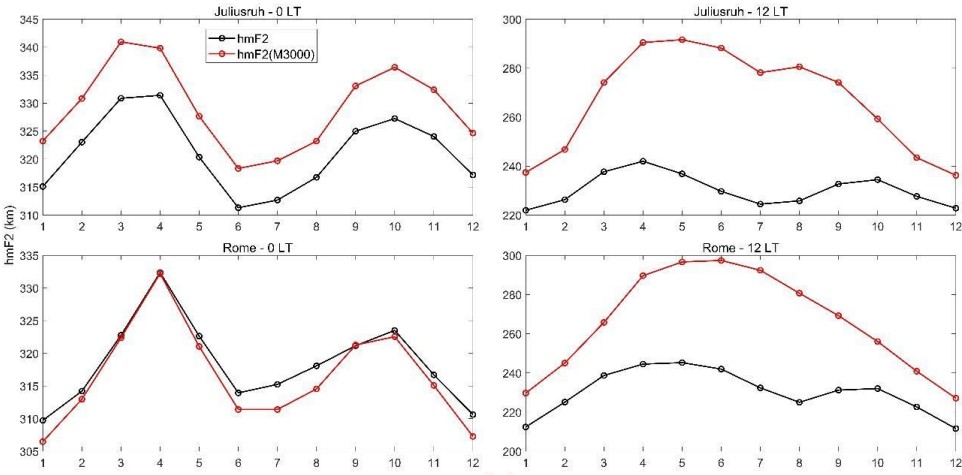


**Figure 1.** hmF2 monthly median average along period 2001-2022 in terms of month, at
Juliusruh (upper panels) and Rome (lower panels), at 0 LT (left panels) and 12 LT (right
panels), considering autoscaled heights (black) and the values obtained using the Shimazaki
formula (red).

However, the trend of the residuals, considering annual means for example, after filtering the
solar activity effect are in good agreement for night and daytime, as can be noticed from
Figure 2.

For this purpose, the simplest filtering was applied, that is considering the residuals of hmF2
from a linear regression with MgII as the EUV solar proxy. There is a general good agreement
in trend values, except in the case of Rome at noon when different signs are obtained between
the autoscaled and Shimazaki hmF2 values. Despite this, we chose to carry out this study
with the Shimazaki formula, given by Equation (1), considering that the errors are systematic
and will not impact the results of the comparative analysis we aim to present. We further
reference the findings of Scotto (2013) to support its use for trend analysis. His results were
obtained for a simulation of nighttime hours with a superimposed trend of −14 km/century
on the hmF2 parameter, which indicate that regardless of the empirical formula used, the
accuracy of hmF2 from ionosonde measurements would be adequate to detect this trend.



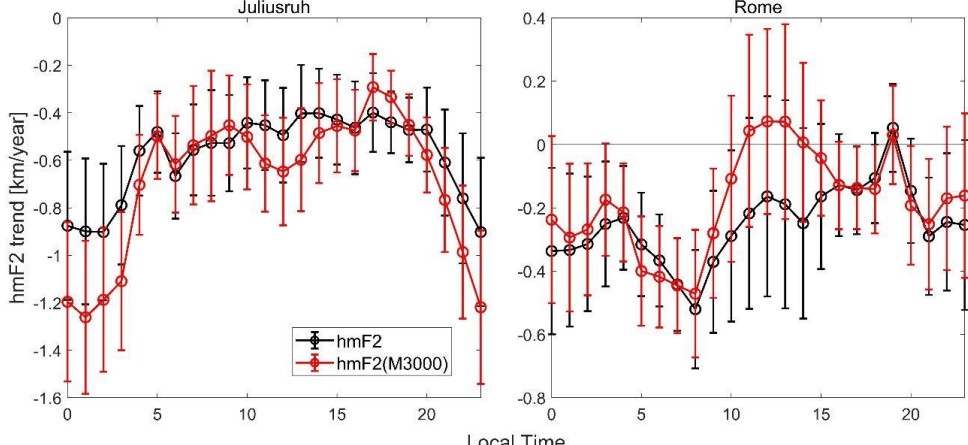

214

**Figure 2.** hmF2 trends (km/year) in terms of local time considering annual means of monthly
median autoscaled heights (black) and the values obtained using the Shimazaki formula (red),
for Juliusruh (left panel) and Rome (right panel), after filtering solar activity using a linear
regression on MgII. The error bars correspond to one standard deviation.

**4. Methodology to compare the different solar EUV proxies and Ap index roles on hmF2 trend analysis**

In order to compare the different solar EUV proxies' effects on the trend estimation process,
we repeat the filtering and trend calculations using each of the five proxies (MgII, F$\alpha$, F10.7,
SN, and F30), which will be generically called X. The filtering, in turn, was performed
considering four models in order to analyze the effect of Ap, which are:

1) Linear regression on X:

$$\text{hmF2} = A + B\,X \tag{2}$$

2) Second degree polynomial regression on X:

$$\text{hmF2} = A + B\,X + C\,X^2 \tag{3}$$

3) Linear regression on X and Ap:

$$\text{hmF2} = A + B\,X + D\,Ap \tag{4}$$

4) Second degree polynomial regression on X and linear on Ap:

$$\text{hmF2} = A + B\,X + C\,X^2 + D\,Ap \tag{5}$$

Thus, the regression variables in each model are: X for 1, X & $X^2$ for 2, X & Ap for 3, and
X, $X^2$ & Ap for 4.



The trend is estimated considering a linear regression of the residuals from these models,
ΔhmF2, and time:
$\Delta hmF2 = [hmF2 - hmF2(modeled)] = \alpha + \beta t$ (6)
In order to determine each solar proxy and Ap suitability for the filtering process, and its
effect on trend values, we considered the squared correlation coefficient, $r^2$, of each of the
four models for each of the five solar proxies together with the values of the linear trend
obtained in each case. A visual comparative analysis is made first by plotting the results
obtained for each variable ($r^2$ and trend values). This is followed by a quantitative comparison
through the estimation of percentage differences considering F30 as the reference EUV solar
proxy, and model 1 as the reference model.
The adjusted $r^2$ value was considered because, in multiple regression, the $r^2$ value increases
as more predictors are added due to the way it is calculated. In contrast, the adjusted $r^2$ value
will decrease if the additional variables do not significantly improve the explanation of the
dependent variables (foF2 and hmF2 in this case).
Concerning $r^2$, the percentage difference to compare the different solar proxies is estimated
as
$100 \times [r^2(X_i) - r^2(F30)]$ (7)
where $X_i$=MgII, Fα, SN or F10.7, using only model 1; while the percentage difference to
compare the different models is estimated as
$100 \times [r^2(model\ i) - r^2(model\ 1)]$ (8)
for model i from model 2 to model 4 using only F30 as the solar proxy.
The same applies to trend values, but relative percentage differences were assessed in this
case, estimated as
$100 \times [\beta (X_i) - \beta (F30)] / \beta (F30)$ (9)
and
$100 \times [\beta (model\ i) - \beta (model\ 1)] / \beta (model\ 1)$ (10)
This analysis is repeated for foF2 to compare the effects of solar proxies and the inclusion of
Ap. Since the study is based on a similar analysis made by Laštovička (2021b, c) who
considered the period 1976-2014, each calculation was also made for this period, and for
1976-2022 that is Laštovička's period updated to 2022.

**5. Results**
Figures 3 and 4 present $r^2$ for each model, at 0 and 12 LT respectively, in terms of each solar
proxy, considering hmF2 and foF2 measured at Juliusruh. Figures 5 and 6 show the




equivalent results for hmF2 and foF2 measured at Rome. It is easily noticed that the longest
period analyzed, 1960-2022, shows the greatest variations in $r^2$ between each solar proxy,
with an improved correlation in the case of SN followed by F10.7 for all the models, at
midnight and noon, which nevertheless does not mean that should be considered the best
proxies (Laštovička, 2024; Zossi et al., 2024). For the shorter periods, particularly excluding
solar cycles 20 and 21, the difference in $r^2$ values is smoothed and MgII emerge as the highest
correlated proxy for most of the cases.
Looking at the same figures, when comparing the different models in hmF2 case, the addition
of variables to model 1 improves the correlation, in particular when Ap is added, something
that in foF2 case is almost not noticed. We can argue that this is because there is more
potential for improvement in hmF2 compared to foF2, as the $r^2$ value is, on average, lower
for hmF2.

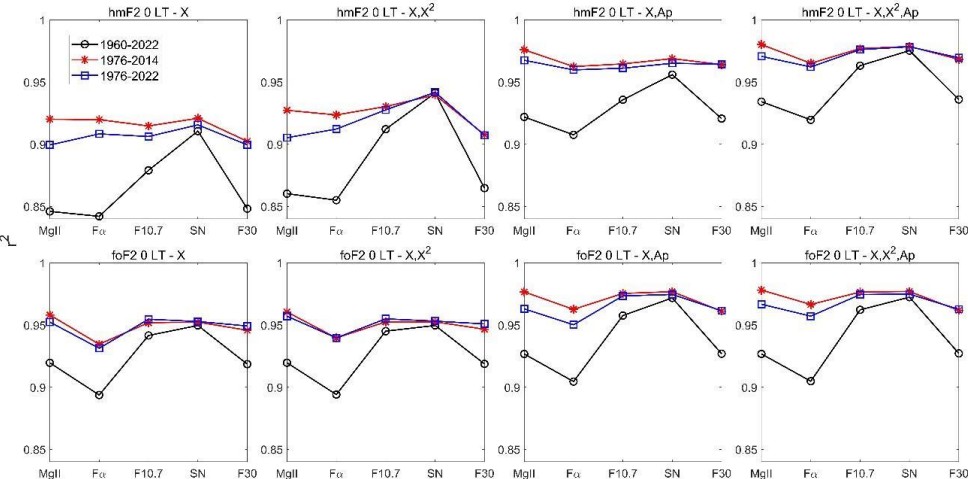


**Figure 3.** Squared correlation coefficient, $r^2$, of hmF2 (upper panels) and foF2 (lower panels)
at 0 LT measured at Juliusruh, within each model (indicated at the top of each panel) in terms
of each solar proxy (MgII, Fα, F10.7, SN and F30). Time series period: 1960-2022 (black),
1976-2014 (red), 1976-2022 (blue).

288





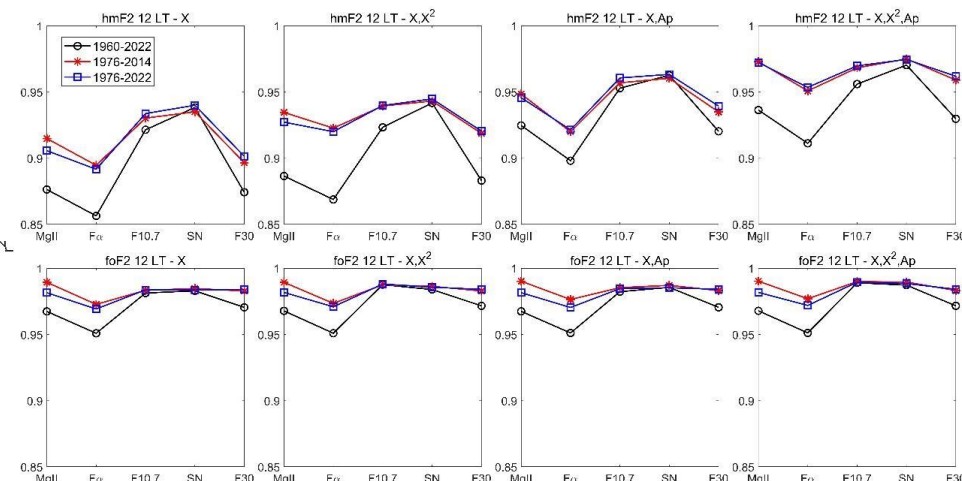

**Figure 4.** Squared correlation coefficient, $r^2$, of hmF2 (upper panels) and foF2 (lower panels) at 12 LT measured at Juliusruh, within each model (indicated at the top of each panel) in terms of each solar proxy (MgII, F$\alpha$, F10.7, SN and F30). Time series period: 1960-2022 (black), 1976-2014 (red), 1976-2022 (blue).

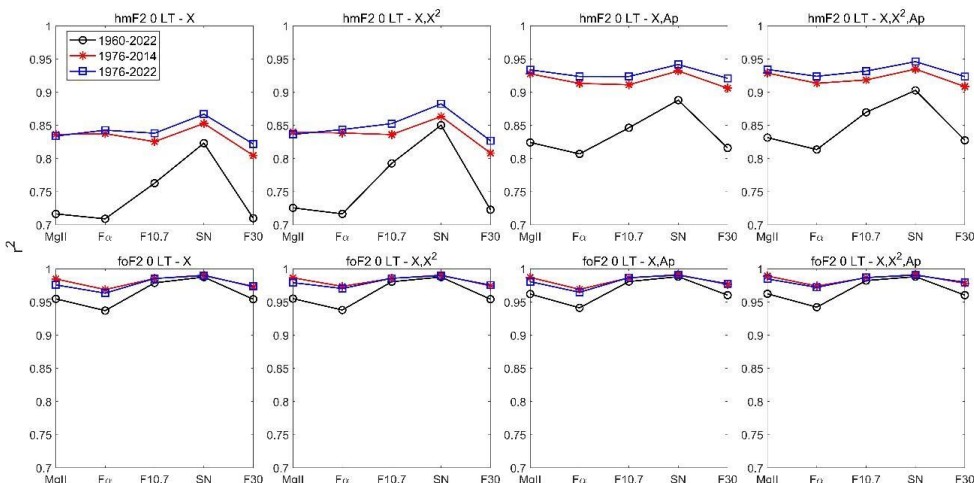

**Figure 5.** Squared correlation coefficient, $r^2$, of hmF2 (upper panels) and foF2 (lower panels) at 0 LT measured at Rome, within each model (indicated at the top of each panel) in terms of each solar proxy (MgII, F$\alpha$, F10.7, SN and F30). Time series period: 1960-2022 (black), 1976-2014 (red), 1976-2022 (blue).





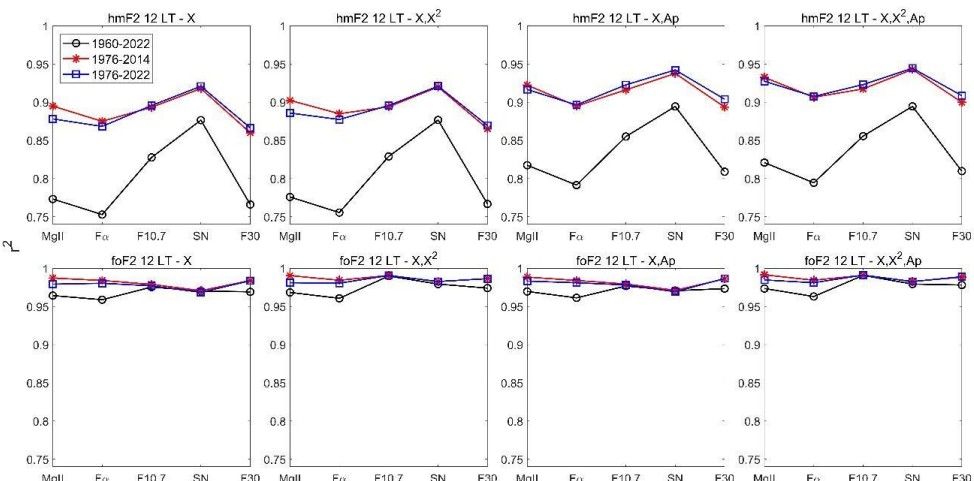

**Figure 6.** Squared correlation coefficient, $r^2$, of hmF2 (upper panels) and foF2 (lower panels) at 12 LT measured at Rome, within each model (indicated at the top of each panel) in terms of each solar proxy (MgII, F$\alpha$, F10.7, SN and F30). Time series period: 1960-2022 (black), 1976-2014 (red), 1976-2022 (blue).

Figures 7 and 8 present trend values obtained after filtering through each of the four models, at 0 and 12 LT respectively, in terms of each solar proxy, of hmF2 and foF2 measured at Juliusruh. Figures 9 and 10 show the equivalent results for hmF2 and foF2 measured at Rome. Similar to foF2 case, hmF2 trends are less negative when the solar proxy used is SN, followed by F10.7. They are more negative when F30, MgII and F$\alpha$ is used instead. In hmF2 case also, the trends get less negative and closer to zero when Ap is included in the model, which is something expected due to the increase obtained in $r^2$. foF2 trends are almost identical with or without Ap included, which is in agreement with the results of other authors showing that Ap do not make a significant difference if included in the filtering process (Laštovička, 2021a). It is worth noting that in hmF2 case there are almost no positive trends except two exceptions: Juliusruh at 0 LT using SN as a proxy in model 3, for periods 1976-2014 and 1976-2022. While in foF2 case, positive trends are obtained for several cases all of which use SN or F10.7 as the solar proxy.





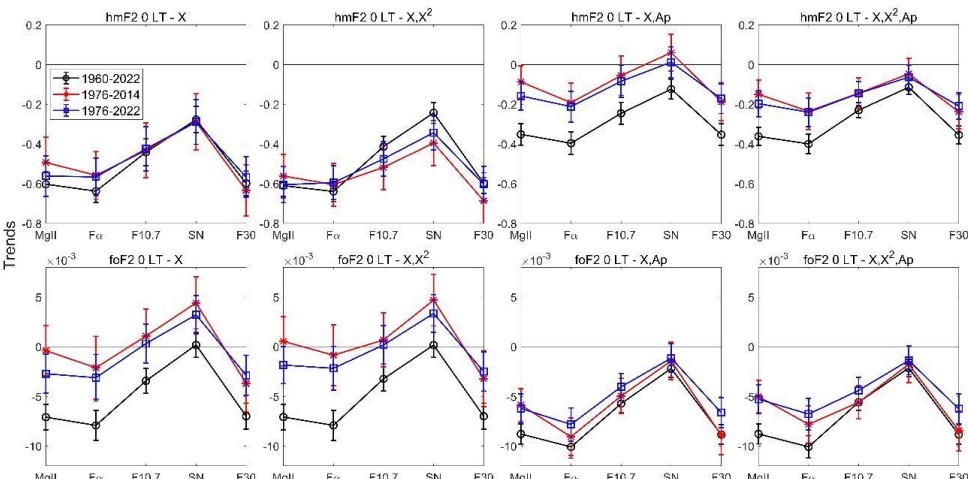

**Figure 7.** Linear trend of hmF2 (upper panels) and foF2 (lower panels) at 0 LT measured at Juliusruh, considering residuals filtered with each model (indicated at the top of each panel) in terms of each solar proxy (MgII, F$\alpha$, F10.7, SN and F30). Time series period: 1960-2022 (black), 1976-2014 (red), 1976-2022 (blue).

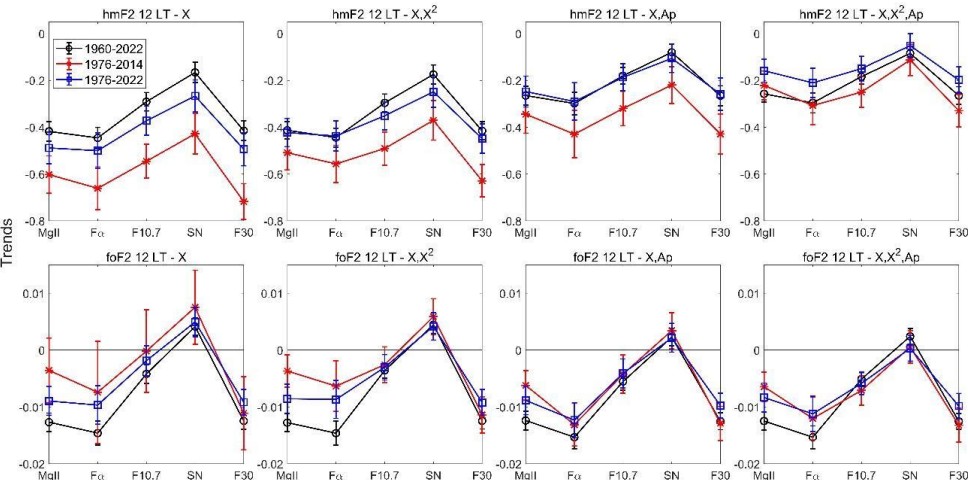

**Figure 8.** Linear trend of hmF2 (upper panels) and foF2 (lower panels) at 12 LT measured at Juliusruh, considering residuals filtered with each model (indicated at the top of each panel) in terms of each solar proxy (MgII, F$\alpha$, F10.7, SN and F30). Time series period: 1960-2022 (black), 1976-2014 (red), 1976-2022 (blue).





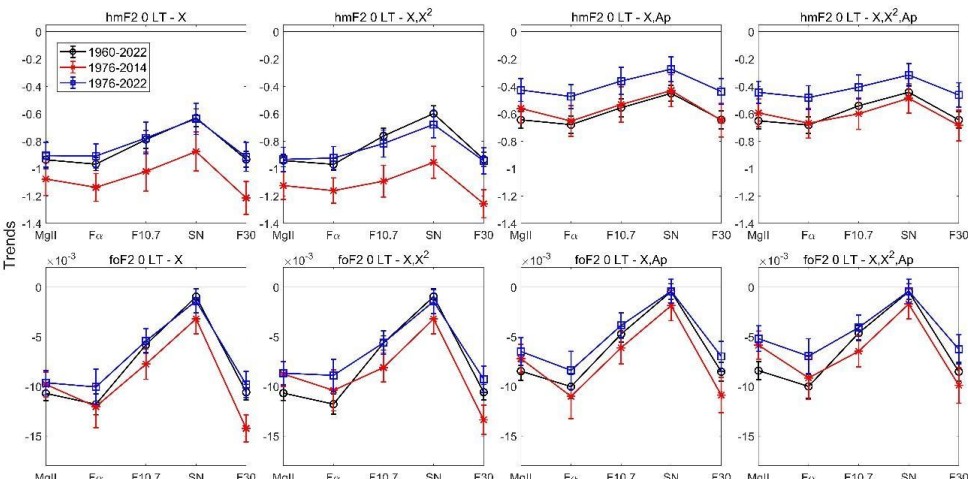

**Figure 9.** Linear trend of hmF2 (upper panels) and foF2 (lower panels) at 0 LT measured at Rome, considering residuals filtered with each model (indicated at the top of each panel) in terms of each solar proxy (MgII, Fα, F10.7, SN and F30). Time series period: 1960-2022 (black), 1976-2014 (red), 1976-2022 (blue).

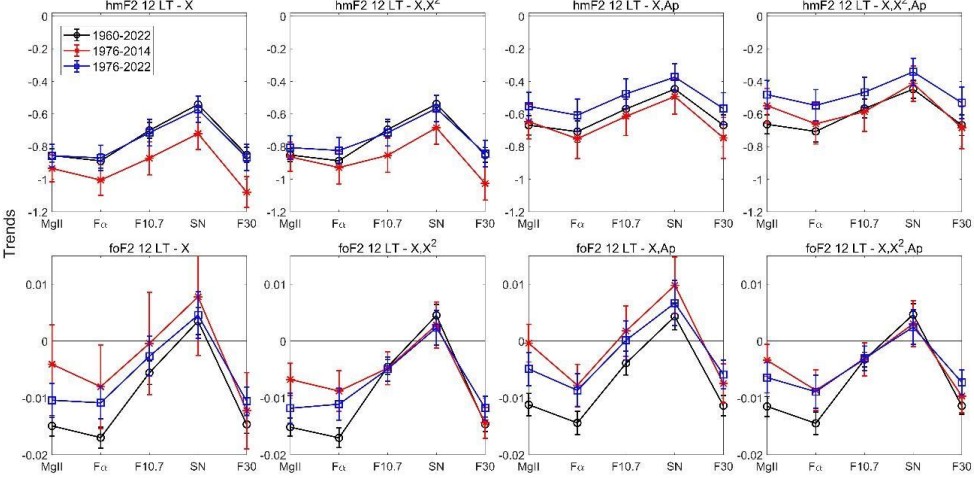

**Figure 10.** Linear trend of hmF2 (upper panels) and foF2 (lower panels) at 12 LT measured at Rome, considering residuals filtered with each model (indicated at the top of each panel) in terms of each solar proxy (MgII, Fα, F10.7, SN and F30). Time series period: 1960-2022 (black), 1976-2014 (red), 1976-2022 (blue).

In order to have a more quantitative analysis of the differences of each solar proxy and of Ap role on filtering we estimated $r^2$ and trend differences with respect to proxies and also to



models as explained in Section 4. We do not show the case of SN in order to simplify the
figures, since its difference is highly notorious just from the Figures 3 to 10.
Figures 11 to 14 show the percentage difference in r$^2$ together with the relative percentage
difference in trends when comparing F30 with each of the other proxies—MgII, Fα, and
F10.7—for both hmF2 and foF2, for each station and local time.
In the case of r$^2$ percentage difference, a positive value means a higher correlation, while a
negative value a lower one. In general, and leaving SN out of discussion in this point, F10.7
is the proxy that mostly improves r$^2$ considering the two stations, both local times, and the
three periods. The are also cases of improvement when considering MgII. Again, we
highlight that this result does not imply a better performance of F10.7 and/or MgII
(Laštovička, 2024; Zossi et al., 2024).
In the case of the trend relative percentage differences, considering that the reference trend
is always negative, a positive value implies a less negative trend or even positive, while a
negative value indicates a more negative one. For the period 1960-2022, trend values are
similar either using F30 or MgII in hmF2 and foF2 cases, while in the shortest period 1976-
2014, F30 gives clearly the most negative trends in all the cases, with strongest effect in foF2.

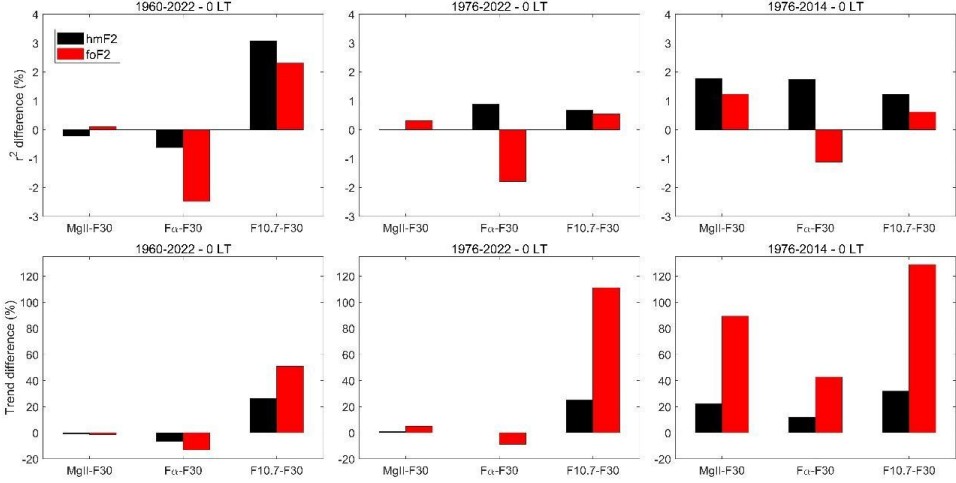


**Figure 11.** r$^2$ percentage difference (upper panels) and trends relative percentage difference
(lower panels), using model 1, between MgII, Fα or F10.7 and F30 for hmF2 (black bars)
and foF2 (red bars) measured at Juliusruh at 0 LT, considering periods 1960-2022, 1976-
2022, and 1976-2014, indicated at the top of each panel.






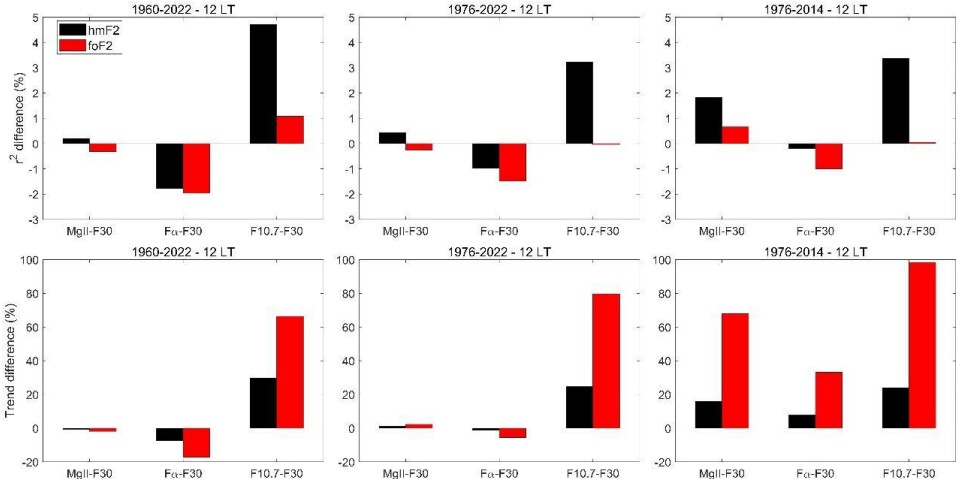

**Figure 12.** r² percentage difference (upper panels) and trends relative percentage difference
(lower panels), using model 1, between MgII, Fα or F10.7 and F30 for hmF2 (black bars)
and foF2 (red bars) measured at Juliusruh at 12 LT, considering periods 1960-2022, 1976-
2022, and 1976-2014, indicated at the top of each panel.

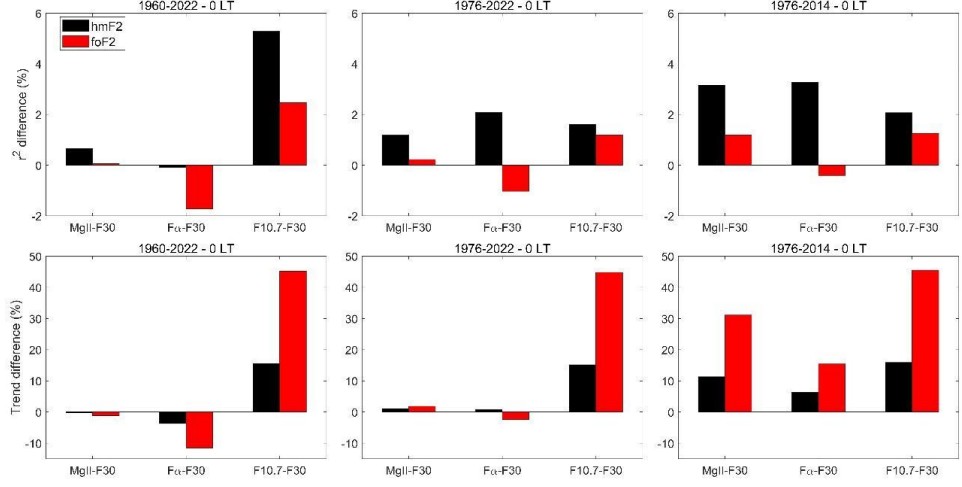


**Figure 13.** r² percentage difference (upper panels) and trends relative percentage difference
(lower panels), using model 1, between MgII, Fα or F10.7 and F30 for hmF2 (black bars)
and foF2 (red bars) measured at Rome at 0 LT, considering periods 1960-2022, 1976-2022,
and 1976-2014, indicated at the top of each panel.





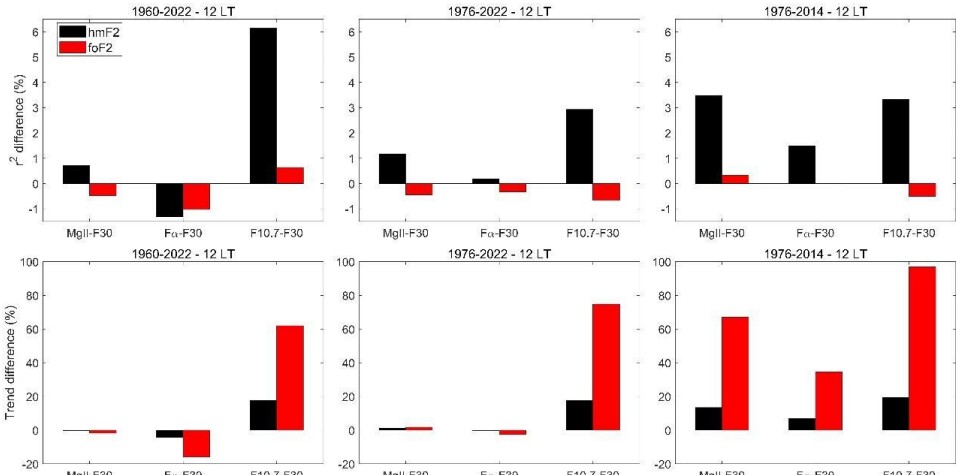


**Figure 14.** $r^2$ percentage difference (upper panels) and trends relative percentage difference
(lower panels), using model 1, between MgII, Fα or F10.7 and F30 for hmF2 (black bars)
and foF2 (red bars) measured at Rome at 12 LT, considering periods 1960-2022, 1976-2022,
and 1976-2014, indicated at the top of each panel.

386

Figures 15 to 18 show the percentage difference in $r^2$ together with the relative percentage
difference in trends when comparing model 1 with each of the other models, for both hmF2
and foF2, at each station and local time. $r^2$ differences are consistently greater for hmF2
compared to foF2 in all cases, meaning that adding the squared solar proxy term and/or the
Ap index always improve the model. Once more, this is statistically reasonable, since hmF2
has a larger margin for improvement. When a model, like that for foF2, already exhibits a
high degree of correlation, incorporating additional variables is less likely to result in
significant improvements. For example, at Juliusruh at 12 LT, neither the Ap index nor the
squared proxy term significantly enhances the foF2 model. This outcome is expected because
maximum solar activity levels typically do not surpass the saturation level, limiting
improvements in correlation for both ionospheric parameters.

In the case of the trend values, again the square term alone does not produce big differences,
while Ap weakens in the negative trends in all the cases except for one: foF2 at Juliusruh, 0
LT.

401



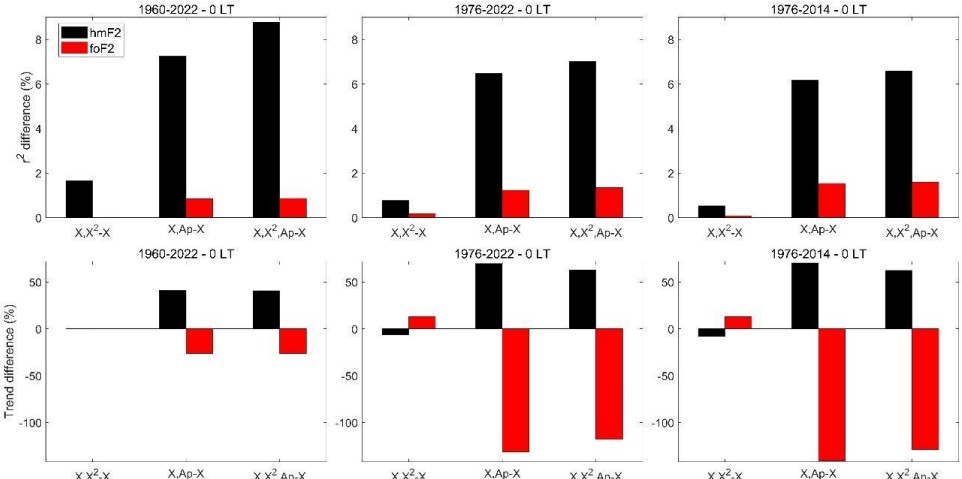

402

**Figure 15.** $r^2$ percentage difference (upper panels) and trends relative percentage difference (lower panels), using F30 as a solr proxy, between models 2, 3 or 4 and model 1 for hmF2 (black bars) and foF2 (red bars) measured at Juliusruh at 0 LT, considering periods 1960-2022, 1976-2022, and 1976-2014, indicated at the top of each panel.

407

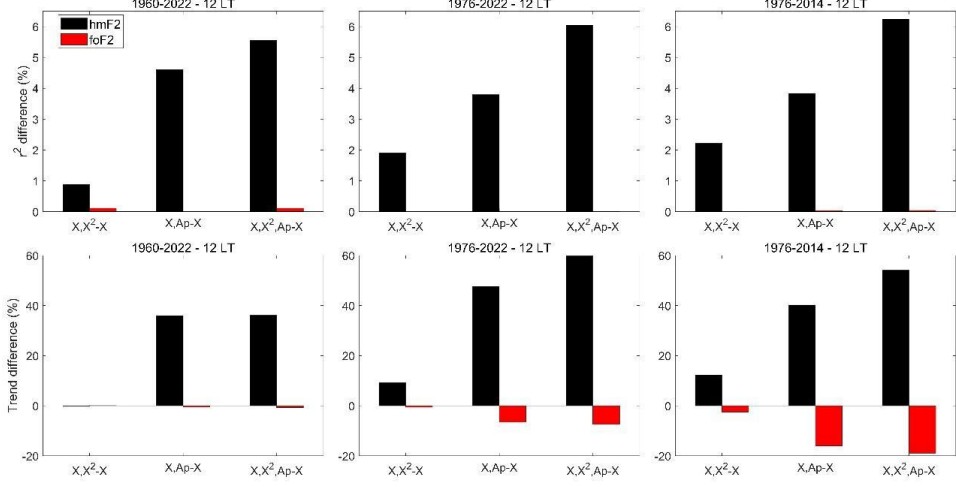

408

**Figure 16.** $r^2$ percentage difference (upper panels) and trends relative percentage difference (lower panels), using F30 as a solar proxy, between models 2, 3 or 4 and model 1 for hmF2 (black bars) and foF2 (red bars) measured at Juliusruh at 12 LT, considering periods 1960-2022, 1976-2022, and 1976-2014, indicated at the top of each panel.



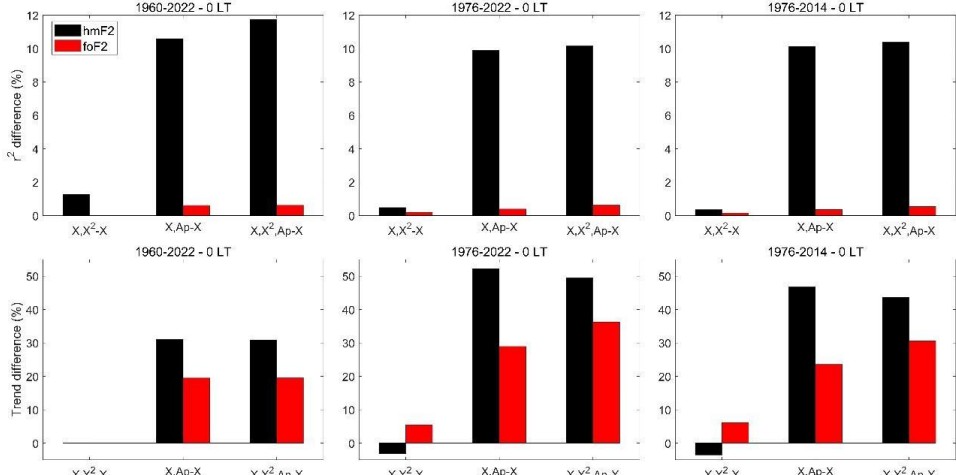

413

**Figure 17.** $r^2$ percentage difference (upper panels) and trends relative percentage difference
(lower panels), using F30 as a solar proxy, between models 2, 3 or 4 and model 1 for hmF2
(black bars) and foF2 (red bars) measured at Rome at 0 LT, considering periods 1960-2022,
1976-2022, and 1976-2014, indicated at the top of each panel.

418

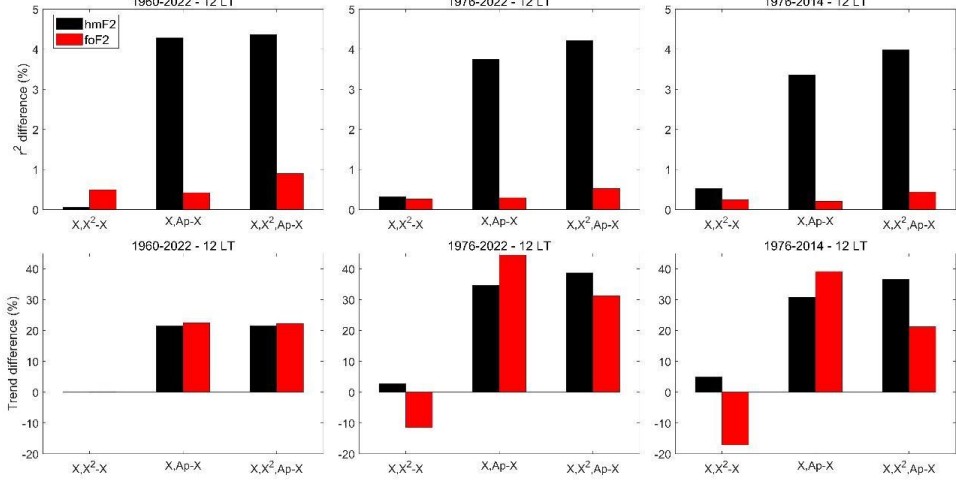

419

**Figure 18.** $r^2$ percentage difference (upper panels) and trends relative percentage difference
(lower panels), using F30 as a solar proxy, between models 2, 3 or 4 and model 1 for hmF2
(black bars) and foF2 (red bars) measured at Rome at 12 LT, considering periods 1960-2022,
1976-2022, and 1976-2014, indicated at the top of each panel.

424

425

426





## 6. Discussion and conclusions

In order to analyze the effect of different solar EUV proxies on hmF2 trend estimation, following the works by Laštovička (2021b, 2021c), we implemented a similar analysis with some additions, to noon and midnight values. Noting that the correlation between hmF2 and solar EUV proxies was systematically lower than in foF2, the inclusion of Ap in the filtering process was incorporated to the analyses.

For both stations, both local times, and the three periods analyzed, $r^2$ values between hmF2 and the solar proxies considering different models which include or not Ap, are consistently lower compared to the corresponding foF2 cases. Thus, the variation in $r^2$ values between different proxies, and between different models are stronger for hmF2, since there is more variance left out to be improved. In contrast, for foF2, the solar proxy linear term typically accounts for almost all the variation, leaving less than 5% of the variance unexplained.

However, with respect to trend values, the difference is more noticeable in foF2 case when comparing different proxies, but not when evaluating the addition or not of Ap. This suggests that foF2 trends seem more sensitive to the proxy used to filter solar activity effect. hmF2 trends are also in general all negative and seem more stable than in foF2 case, probably related to the fact that the greenhouse effect is expected to be more clear in hmF2 than in foF2 (Rishbeth, 1990; Rishbeth and Roble, 1992).

An aspect which deserves further discussion is the comparison of our results between the three periods considered. Differences, in $r^2$ and in trends as well, are more noticeable during the longest period: 1960-2022. This can be explained looking at the long-term variation of each solar proxy that is linked to the Gleissberg cycle, of ~80-100-year quasi-periodicity. Figure 19 highlights this more clearly by displaying the normalized annual mean values of the five proxies here considered, together with the envelope that joins the maximum and minimum values of each solar cycle in the period 1960-2022. The Gleissberg cycle is shown by the maximum values, having the most recent peak in cycle 22 (~1990). The increasing phase of this long-term cycle is clearly observed before cycle 22, followed by the beginning of the decreasing phase. While the well-known ~11-year cycle is quite similar for all the solar proxies, the Gleissberg cycle is not, being SN the index with the greatest differences. It is also clear from this figure that, while longer the period within the 1960-2022 interval, more differences are included since more maximum periods enter into the time series analyzed, and that could explain the stronger differences we found for the period 1960-2022 in comparison to the shorter ones in most of the cases.

A similar effect is produced by differences in the minimum epochs, but in the opposite sense. This is not supposedly part of the Gleissberg cycle, but it is clear that since the 1996 minimum epoch, the following minima present weaker indices' values in all the cases, but with different decreasing levels. Therefore, if the series starts closer to 1996, the trend will be more pronounced than if the time series begins earlier. Consequently, more significant differences should be observed in shorter periods, especially if they include one or both of the recent minima around ~2008 and ~2019.



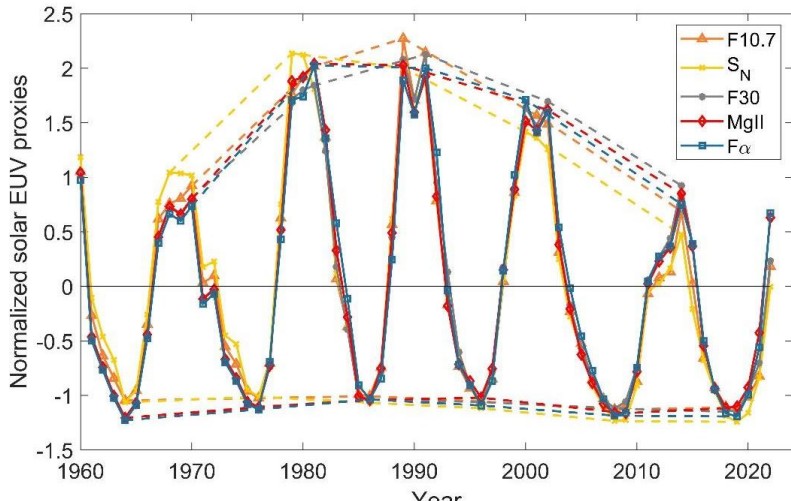

**Figure 19.** MgII (red diamond), Fα (blue square), F10.7 (orange triangle), SN (yellow cross) and F30 (gray dot) normalized annual means (period 1960-2022). Dashed lines join the maximum and minimum values of each solar cycle.

We bring back here Bremer (1992) conclusion where he mentions that an important demand is the correct filtering of the solar and geomagnetic influence on the data because it causes variations that are much larger than the trends of interest. We here emphasize this aspect of trend assessments showing once again that the problem is not yet fully resolved and deserves to be further and more deeply investigated and expanded.

**Statements and Declarations**

The authors have no competing interests to declare that are relevant to the content of this article. Only Ana G. Elias, who is also an author of this work, is a member of the editorial board of Annales Geophysicae.

**Acknowledgements**

T. Duran, Y. Melendi and F. Buezas acknowledge research project PGI 24/J089. A.G. Elias, B.S. Zossi and B.F. de Haro Barbas acknowledge research projects PIUNT E756 and PIP 2957. Also, we acknowledge GIRO data resources http://spase.info/SMWG/Observatory/GIRO.

**Data Availability**

Ionospheric M(3000)F2 and foF2 data for Rome and Juliusruh were obtained from the World Data Centre (WDC) for Space Weather, Australia, accessible at



https://downloads.sws.bom.gov.au/wdc/iondata/au/ and from Damboldt and Suessman
database available in the same WDC
(https://downloads.sws.bom.gov.au/wdc/iondata/medians/). In the case of Rome, to extend
the dataset until 2022, additional data were incorporated from the Digital Ionogram Data
Base (DIDBase) at Lowell GIRO Data Center (LGDC). Juliusruh data is also available from
the Leibniz-Institute of Atmospheric Physics at https://www.ionosonde.iap-
kborn.de/mon_fof2.htm. hmF2 autoscaled values for both stations were obtained from
LGDC. MgII data is obtained from the University of Bremen at https://www.iup.uni-
bremen.de/UVSAT/data/; Hydrogen Lyman α flux is accessible from the LASP Interactive
Solar Irradiance Data Center, University of Colorado, at
https://lasp.colorado.edu/data/timed_see/composite_lya/lyman_alpha_composite.nc; SN
annual mean values were directly obtained from SILSO (Sunspot Index and Long-term Solar
Observations - Royal Observatory of Belgium, Brussels) sourced at
http://www.sidc.be/silso/datafiles; F10.7 series are provided by Space Weather Canada at
https://spaceweather.gc.ca/forecast-prevision/solar-solaire/solarflux/sx-en.php; F30 is
available from the Nobeyama Radio Polarimeters (NoRP) at
https://solar.nro.nao.ac.jp/norp/index.html. Ap index was obtained from the Kyoto World
Data Center for Geomagnetism at https://wdc.kugi.kyoto-u.ac.jp/index.html.

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
