# Peer review of "Impact of different solar EUV proxies and Ap index on hmF2 trend analysis"

_EGUsphere, 2024_

## Referee Comment (RC1)

The paper describes an approach to looking for the hmF2 trends. More or less similar job has been described for the foF2 trends in the previous work of the same group. The principal introducing point of the authors is that there is a need for "…the correct filtering of the solar and geomagnetic influence on the data…" in looking for trends in foF2 and hmF2. They state that "…the problem is not yet fully resolved and deserves to be further and more deeply investigated and expanded."

In the Introduction, the authors briefly describe some previous approaches to looking for the hmF2 trends and present their concept of the problems related to that task.

The ionospheric data used for the analysis are presented in Section 2. The observations at Rome and Juliusruh stations are analyzed. The foF2 and M(3000) parameters are used to calculate hmF2 by Shimazaki formula.

To get rid of SA effects in the hmF2 changes, the authors consider usual set of five solar proxies: F10.7, SN, Ly-a, MgII, and F30. The difference in their approach is that they consider also possible effects of changes in geomagnetic activity.

The authors consider a reliability of the Shimazaki formula. They conclude that it works well at night but overestimates hmF2 in the daytime. However, in terms of trends, the hmF2 values by Shimazaki formula work well, this fact being illustrated visually by Fig. 2.

Various kinds of regression equation used to calculate the hmF2 trend are described in Section 4.

To find effects of each solar proxy and geomagnetic activity on the process that the authors call "filtering", the authors analyze the squared correlation coefficient $r^2$ for all four models and all five solar proxies together with the trend values obtained in each case

The consideration of the $r^2$ values for hmF2 and foF2 over three periods 1960-2022, 1976-2014, and 1976-2022 (Figs. 3-6) shows that inclusion of more complicated regressions (especially with allowance for Ap index) increases the $r^2$ values for hmF2. However, it is not so for foF2.

I think that the most interesting results are presented in Figs. 7-10. They show the hmF2 and foF2 trends for each of three aforementioned periods with "filtering" by various SA proxies. An important point is that in all cases the trends for the longest period of 1960-1922 are the lowest. In my opinion, it is one more proof that trends of anthropogenic origin have begun to appear somewhere after 1980.

The results in $r^2$ and trends for different approaches to the "filtering" by three SA proxies (F10.7, MgII, and Ly-a) as compared to F30 are shown in Figs. 11-18. The results of using more complicated regression than the simplest one show that the $r^2$ differences are substantially greater for hmF2 than for foF2 in all cases. It is an important result demonstrating that the hmF2 trends are more sensitive to inclusion of Ap effects more complicating regressions in the "filtering' than the foF2 trends.

I consider the paper as a very interesting step helping substantially in solving the great problem of deriving long-term trends in parameters of the F2 layer. I recommend publication of the paper as it is. My only small comment is that there are no units of the trends at the ordinates of figs 7-10. I assume that they are km/year and MHz/year, but it should be indicated in the plots or captions.

---

## Author Response (AR1)

**Answer to Reviewers**

**Answer to Reviewer #1:**

Thank you very much for your valuable comments and for considering the results of this ionospheric trend analysis to be important.

Following are our answers (in black) to your comments (in blue).

The changes in the revised manuscript which correspond to your remarks appear in red, together with those corresponding to the comments of Reviewer #2.

In particular, regarding your comment "An important point is that in all cases the trends for the longest period of 1960-1922 are the lowest. In my opinion, it is one more proof that trends of anthropogenic origin have begun to appear somewhere after 1980":

We fully agree with this observation. Additionally, it is worth noting that 1980 coincides with the solar cycle 21 maximum. In terms of the Rz proxy, the maximum solar activity levels began to decline following the peak in 1979-1980, which aligns with the Gleissberg cycle in solar activity maxima. For F30 and F10.7 proxies, this decrease in solar maximum levels began after solar cycle 22, around 1990. Thus, it is probable that a combination of the anthropogenic effect becoming more prominent after 1980 and the decrease in solar maxima since this period contributes to the observed trends. However, this argument depends on the solar activity proxy considered. We explored this point further and added a paragraph about it in the conclusions of the revised version of the manuscript.

We also consider an important result that "hmF2 trends are more sensitive to inclusion of Ap effects more complicating regressions in the 'filtering' than the foF2 trends", so thank you for pointing this out.

About your comment "My only small comment is that there are no units of the trends at the ordinates of figs 7-10. I assume that they are km/year and MHz/year, but it should be indicated in the plots or captions.":

We added the units to the captions of the trends in Figs 7-10. You are correct, they are km/year and MHz/year.
* * *
**Answer to Reviewer #2:**

Thank you very much for your constructive feedback and for considering our manuscript suitable for publication. We appreciate your positive assessment of our work and your suggestions for improvement.

Following are our answers (in black) to your comments (in blue).

The changes in the revised manuscript which correspond to your remarks appear in red, together with those corresponding to the comments of Reviewer #1.

Regarding your specific comments:

- P3L96: 'E107' to 'E10.7': We corrected this typo and ensured that it reads 'E10.7' as suggested.

- P14L355: 'The are': We fixed this grammatical error.

- P20, Figure 19: In legend replace $S_{N}$ to 'SN': We adjusted the legend in Figure 19 and replaced '$S_{N}$' with 'SN' for consistency and clarity.

- I suggest enlarging the title and the text of the X and Y axes to make them more readable: We agree that improving the readability of the figures is important. We enlarged the titles and the text of the X and Y axes in all relevant figures.

Thank you again for your suggestions.

Hoping to meet all your requirements,

Trinidad Duran, Bruno S. Zossi, Yamila D. Melendi, Blas F. de Haro Barbas, Fernando S. Buezas, & Ana G. Elias